# An In-Depth Assessment of the Drivers Changing China's Crop Production Using an LMDI Decomposition Approach

Yuqiao Long [1,2], Wenbin Wu [1,*], Joost Wellens [2], Gilles Colinet [2] and Jeroen Meersmans [2]

1   Key Laboratory of Agricultural Remote Sensing (AGRIRS), Ministry of Agriculture and Rural Affairs/Institute of Agricultural Resources and Regional Planning, Chinese Academy of Agricultural Sciences, Beijing 100081, China
2   TERRA Teaching and Research Centre, Gembloux Agro-Bio Tech, University of Liège, 5030 Gembloux, Belgium
*   Correspondence: wuwenbin@caas.cn

**Abstract:** Over the last decades, growing crop production across China has had far-reaching consequences for both the environment and human welfare. One of the emerging questions is "how to meet the growing food demand in China?" In essence, the consensus is that the best way forward would be to increase crop yield rather than further extend the current cropland area. However, assessing progress in crop production is challenging as it is driven by multiple factors. To date, there are no studies to determine how multiple factors affect the crop production increase, considering both intensive farming (using yield and multiple cropping index) and large-scale farming (using mean parcel size and number of parcels). Using the Logarithmic-Mean-Divisia-Index (LMDI) decomposition method combined with statistical data and land cover data (GlobeLand30), we assess the contribution of intensive farming and large-scale farming changes to crop production dynamics at the national and county scale. Despite a negative contribution from *MPS* (mean parcel size, −25%), national crop production increased due to positive contributions from yield (+77%), *MCI* (multiple cropping index, +27%), as well as NP (number of parcels, +21%). This allowed China to meet the growing national crop demand. We further find that large differences across regions persist over time. For most counties, the increase in crop production is a consequence of improved yields. However, in the North China Plain, NP is another important factor leading to crop production improvement. On the other hand, regions witnessing a decrease in crop production (e.g., the southeast coastal area of China) were characterized by a remarkable decrease in yield and *MCI*. Our detailed analyses of crop production provide accurate estimates and therefore can guide policymakers in addressing food security issues. Specifically, besides stabilizing yield and maintaining the total NP, it would be advantageous for crop production to increase the mean parcel size and *MCI* through land consolidation and financial assistance for land transfer and advanced agricultural infrastructure.

**Keywords:** LMDI; drivers; yield; crop production change; China

---

## 1. Introduction

Increases in crop production over the last half-century due to cropland expansion and technological advances (e.g., fertilizers, pesticides, irrigation, etc.) have had far-reaching consequences for human welfare and the environment [1]. However, intensive farming has resulted in a large scale conversion of grassland areas into croplands, adversely affecting biodiversity [2–4] and climate change [5]. Unfortunately, these threats will probably intensify as projections estimate an increase in crop demand by at least another 50% by 2050 [6]. These impacts are likely to be worse for developing countries with a large population, such as China.

China has achieved remarkable crop production growth over the first decade of the 21st century, with total production equal to 552 million tons in 2010 according to





records from the Food and Agriculture Organization of the United Nations (FAO) (https://www.fao.org/faostat/en/#data, accessed on 14 December 2021). In particular, the continuous growth of crop production between 2003 and 2010 has been important in delivering one of the most essential ecosystem services, i.e., food security. However, it is expected that China's food demand will continue to rise in the coming decades, as the country is increasingly dependent on imported food and feed [7]. Additionally, during this period some land use change and management issues affecting food supply have occurred, including natural disasters [8], decline of high-quality cropland [9], and cropland abandonment [10]. In order to provide theoretical support for agricultural policymaking, it is necessary to clarify the extent to which different factors are driving the change in China's crop production.

Since 1961, the "Green Revolution" associated with the introduction of new agricultural technologies and practices has greatly improved China's crop production. An increase of 385% has been seen despite the fact that the overall cropland area only expanded by 29% according to the Yearly Statistics Book of the National Bureau of Statistics of China (NBSC) and FAO (http://www.stats.gov.cn/tjsj/ (accessed on 14 December 2021) and https://www.fao.org/faostat/en/#data (accessed on 14 December 2021)). Crop production can be defined as the total output of crop yield per unit area per year multiplied by its area. Although this metric has been widely used to quantify changes in agricultural production, it needs to be recognized that a large set of factors may have an impact on production, including several factors that have remarkably different effects on farming practices and technological advancements, altering entire food production systems [1].

In the literature, crop production across China has typically not been studied in a simultaneously comprehensive and systematic way due to complicated agricultural settings. First, a thorough understanding of the spatial-temporal dynamics of crop production is hampered by the fact that most studies so far have focused on individual factors in isolation, such as drip irrigation [11], consumption of agricultural chemicals [12,13], policy regimes [13,14], or have just analyzed crop production as a whole in which individual factors are lumped together [15]. Second, in these studies the influencing factors of crop production are hierarchical, for example, multiple cropping index, yield, mean parcel size, and number of parcels have direct impacts on crop production, whereas the above factors (e.g., drip irrigation) have an indirect impact on crop production by affecting the four directly influencing factors (i.e., multiple cropping index, yield, mean parcel size, and number of parcels). Hence, by assigning all types of components at the same influencing level, and therefore neglecting the interaction of influencing factors among the different levels, it is not possible to identify the true drivers behind crop production [16]. Third, until now the Logarithmic-Mean-Divisia-Index (LMDI) has mainly been studied at the national and provincial scale [17–20], which is not detailed enough to reveal the underlying driving factors often operating at a smaller spatial scale. In fact, counties are the fundamental units of crop production in China, because the drivers within a given province may vary significantly at the county level [21]. Hence, obtaining an understanding of the spatio-temporal variability in crop production and its associated drivers at the county scale is crucial for the development of regional policies to combat food security issues and to address the associated United Nations Sustainable Development Goals (SDGs). Fourth, the lack of studies that cover the entire nation makes it difficult to make interregional comparisons. As a solution to these critical shortcomings, we introduce a framework that decomposes crop production into factors, which can all be quantified.

Structure Decomposition Analysis (SDA) and Index Decomposition Analysis (IDA) are two commonly used methods for the decomposition of indicator changes. The IDA methodology is easy to use because of its simplicity and flexibility, compared to the SDA method, therefore IDA has been extensively applied. IDA can be further divided into the Divisia Index Decomposition technique and Laspeyres Index Decomposition technique [22]. We applied the Logarithmic-Mean-Divisia-Index method (LMDI), which has many advantages, including perfect decomposition (no residuals), zero-value robustness, time-reversal

symmetry, and consistency in aggregation [23,24]. The LMDI has been widely applied in industrial research focusing on energy efficiency and carbon dioxide emissions at the national scale [25–27]. However, to our knowledge, very few studies have utilized this decomposition methodology to unravel factors influencing crop production.

The first decomposition factor is yield, which is the output of a given crop per unit of harvested cropland area for a given location, typically expressed in tons/ha. The second factor is the multiple cropping index (*MCI*), the average frequency at which crops are planted per year or accumulation of harvest area on a given unit of cropland area per year. Yield and *MCI* are both a clear indicator of intensive farming, but they are nonetheless important to distinguish as they do not necessarily reflect the same aspects of crop production [28].

The next two factors are related to large-scale farming [29,30]. We consider two important landscape metrics, i.e., the number of parcels (NP) and mean parcel size (*MPS*), in order to reflect how farmers cultivate as a function of cropland size, which indicates the degree of large-scale farming at a given location. An increase in large-scale farming can affect crop production in the sense that a shift toward lower farming costs could result in higher farming returns, and hence, higher crop production [14,31]. Knowing how large-scale farming changes is therefore critical to understanding its impact on ongoing food security issues. However, this critical research question has received little attention from researchers so far.

Hence, this study aims to: (1) decompose crop production into relative factors (*yield*, *MCI*, *MPS*, and *NP*) and map the change of each factor between 2000 and 2010 using GlobeLand30 and statistics data; (2) assess the contribution of *yield*, *MCI*, *MPS*, and *NP* to crop production changes at both the nation and county scale; and (3) map the spatial pattern of these contributions at the county scale and identify the dominant factors.

By applying a novel graphic information system (GIS) based approach, we were able to produce a detailed spatial-temporal analysis of how agriculture has met the rising food demand over the first decade of the 21st century.

## 2. Materials and Methods

### 2.1. Study Area

In this study, we consider all of China. The complex topography and, hence, wide range of climates has resulted in many different cropping systems throughout China (Figure 1). For example, in some parts of central China and northern China, only a single rice crop is grown each year because the growing season is too short for multiple crops, whereas in the south and some parts of central China, farmers practice a double rice cropping system [32]. When considering hydrothermal conditions of the landscape, for the same crops and altitudes higher grain yields are obtained on slopes orientated towards the south compared to other aspects, because of warmer temperatures for these topographical positions. However, as a result of steep mountainous terrain, farmers cultivate land everywhere, which has led to considerable fragmentation of the cropland area, and therefore gradually decreased the average field size. This is phenomena is most remarkable in south China.

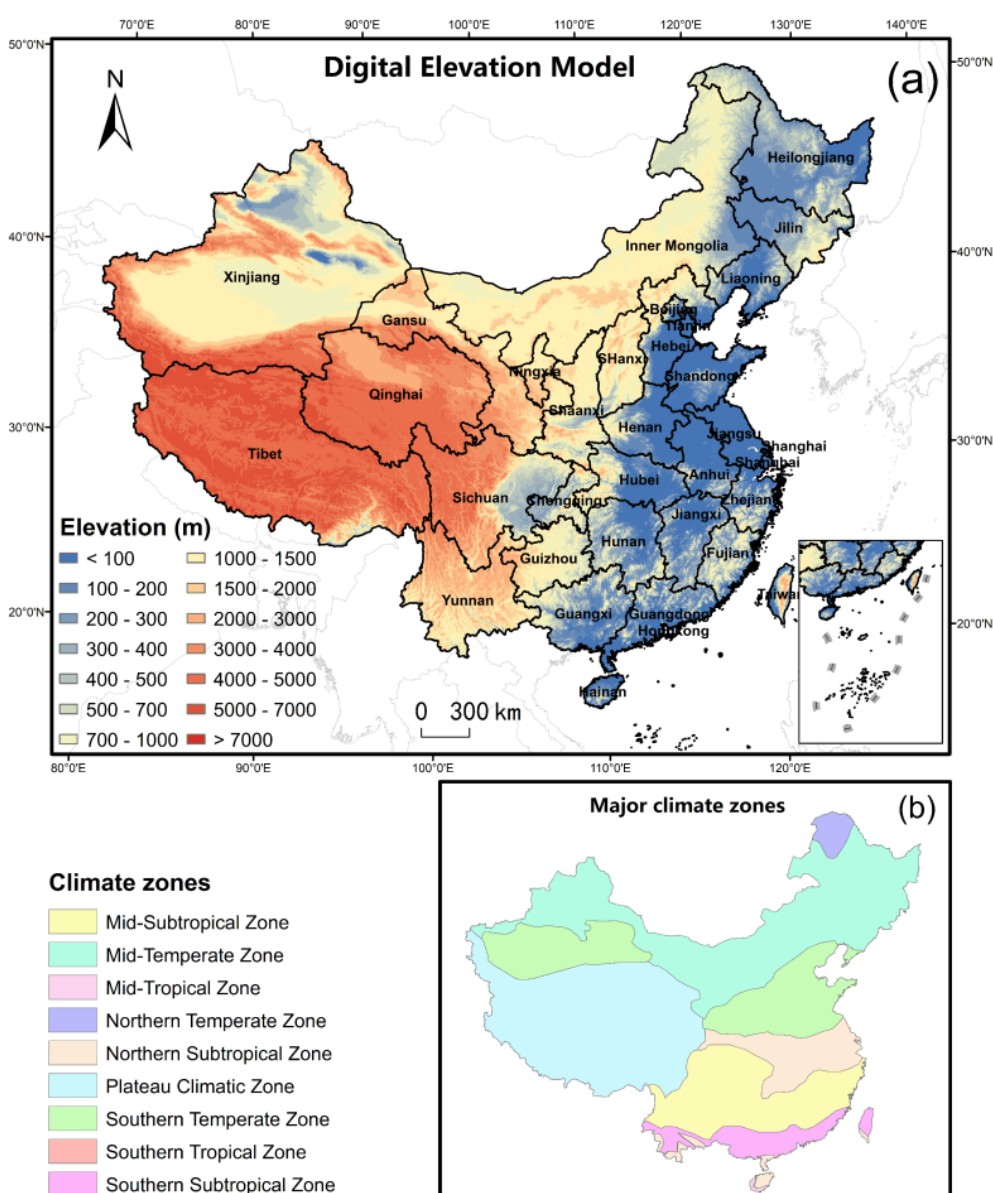

**Figure 1.** Digital elevation model (**a**) and climate zones of China (**b**) with annotation of the provinces.

*2.2. Data Sources*

    An overview of the data used in this paper is shown in Table 1. Spatial and statistical data for 2000 and 2010 were collected. Our analysis included 2396 counties across China for which data on yield and *MCI* were available from the China County Statistical Yearbook (NBSC, 2001 and 2011). The county administrative boundary was acquired from the Geographical Information Monitoring Cloud Platform (http://www.dsac.cn/, accessed on 1 December 2020). The administrative divisions of cities were adjusted during the studied periods. Consequently, the administrative boundaries for 2010 were adopted to make the county unit uniform in size. However, in this study, we did not consider Taiwan, Hong Kong, and Macao due to data unavailability. Data on agriculture was obtained from the NBS. To calculate yield and *MCI*, the total crop production and harvest area were collected from China Agricultural Statistical Yearbook. To reduce the impact of uncertainty in the census data, missing data were complemented using the three-year average data.

**Table 1.** Data used in this paper.

| Data Type | Year | Description | Data Sources |
|---|---|---|---|
| Agricultural data | 2000, 2010 | Crop production in each county Cropland harvest area in each county | China County Statistical Yearbook, NBSC, 2001 and 2011 |
| Land-use imagery | 2000, 2010 | GlobeLand30 land-cover dataset with 30 m spatial resolution across China | www.globallandcover. com (accessed on 1 December 2020) |
| Auxiliary data | 2010 | County administrative boundary in China | http://www.dsac.cn/ (accessed on 1 December 2020) |

In our study, we used GlobeLand30 (Figure 2), which is the world's first global land cover dataset at a resolution of 30 m (http://www.globallandcover.com, accessed on 1 December 2020)). The GlobeLand30 dataset was extracted from more than 20,000 Landsat and Chinese HJ-1 satellite images and based on the integration of pixel- and object-based methods with knowledge (POK-based) [33,34]. This product consists of 10 land cover types among which cropland was defined as the agricultural land use type, including dry land, paddy fields, artificial grasslands, tea plantations, greenhouses, fallow land, and even abandoned cropland [35]. The overall accuracy of the GlobeLand30 derived cropland area across China is higher than 80% [36–38]. In addition, we found that the cropland area of GlobeLand30 is consistent with statistical data at the county scale (see Supplementary Data Figure S1). This relatively high spatial resolution and wide spatial coverage provides us with the opportunity to assess some key factors characterizing large-scale farming, i.e., number of parcels (NP) and mean parcel size (*MPS*) across China using the FRAGSTATS 4.2 software [39].

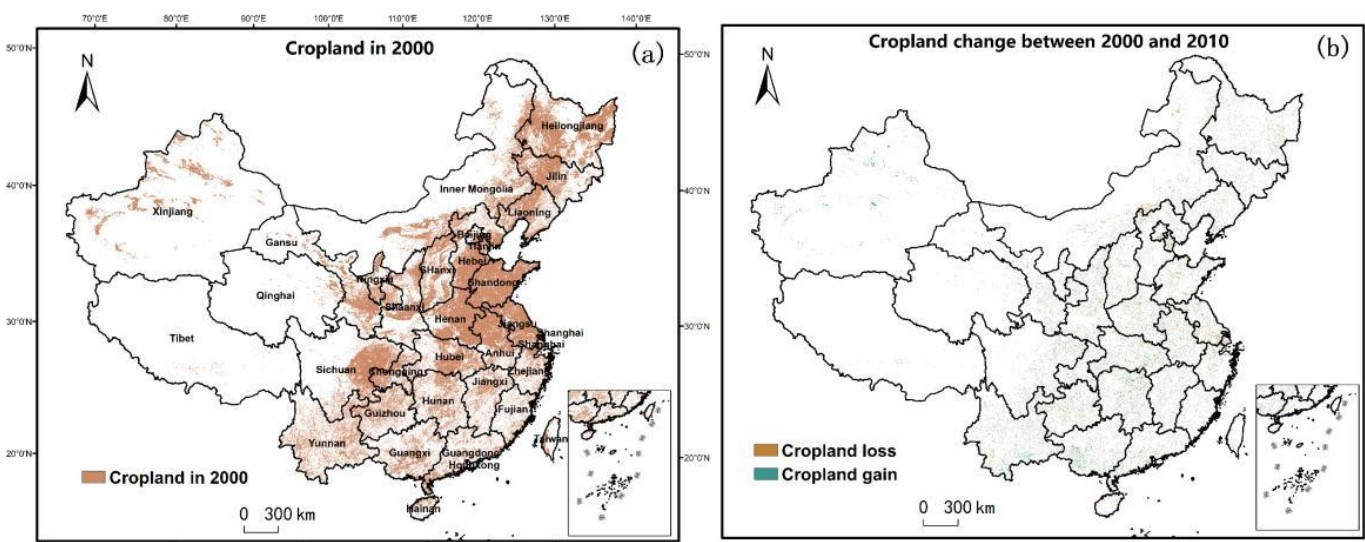

**Figure 2.** Spatial pattern of cropland in GlobeLand30 ((**a**): 2000, (**b**): 2000–2010). Note that the overall accuracy of GlobeLand30 in China has reached 80%, including the cropland layer. The maps were generated using ArcGIS 10.2 ESRI (Environmental Systems Research Institute, 2013).

*2.3. Methods*

2.3.1. Yield and *MCI* Estimation

Two significant indicators that are used to quantify the input and output of cropland in a detailed way are crop yield and *MCI* (Equations (1) and (2)), which are used to reflect the degree of intensive farming.

$$yield = C_{total}/H \tag{1}$$

$$MCI = H/A \tag{2}$$

where *yield* (tons/ha) represents crop production level, $C_{total}$ (tons) is the total crop production in a given region, *H* (ha) is the total harvest area in a given region, *MCI* represents crop production frequency, and *A* is total cropland area in a given region (ha).

### 2.3.2. *MPS* and NP Estimation

Landscape metrics are quantitative measures used to characterize landscape patterns. These metrics are used to establish associations between landscape patterns and landscape processes. In this study, we calculated two landscape metrics (i.e., *NP* and *MPS*) to assess the degree of large-scale farming. To do so, we used FRAGSTATS 4.2 software, which allowed us to calculate these indicators directly from the spatial pattern of cropland. When large-scale farming develops positively, the number of parcels may decrease whereas the mean parcel size increases. This means that the spatial distribution of cropland becomes more concentrated. Hence, by using these indicators, we can evaluate the degree of large-scale farming in each county and the entire nation considering the period 2000–2010.

### 2.3.3. LMDI Decomposition

To analyze the changes of crop production over a given period of time considering multiple factors, we adopted a LMDI analysis [23]. As crop production in a certain region has been shown to be highly related to *yield*, MCI, and cropland area (*A*), we considered these factors in our analysis. Cropland area (*A*) was further divided into *NP* and *MPS*.

Our analysis is based on crop production in which $C_{total}$ is the sum of crop production expressed in tonnes per year, which was calculated as follows:

$$C_{total} = \sum_{i=1}^{n} C_i \tag{3}$$

where $C_i$ is the crop production of county *i*.

$$C_{total} = \sum_{i=1}^{n} yield_i.MCI_i.A_i = \sum_{i=1}^{n} \frac{C_i}{H_i}.\frac{H_i}{A_i}.\frac{A_i}{NP}.\text{NP} = \sum_{i=1}^{n} yield_i.MCI_i.MPS_i.NP_i \tag{4}$$

Subsequently, the $C_{total}$ was disaggregated into four factors, i.e., yield ($yield_i = \frac{C_i}{H_i}$, *tons/ha*), MCI ($MCI_i = H_i/A_i$), MPS ($MPS_i = \frac{A_i}{NP}$, *ha*), and $NP_i$.

The LMDI was divided into additive decomposition and multiplicative decomposition. However, the two decomposition methods display similar results. Because the additive decomposition approach is easier to use and interpret compared to the multiplicative decomposition approach, the following additive decomposition was employed:

$$\Delta C_{total} = C^T{}_{total^i} - C^t{}_{total^i} = \Delta C_{yield^i}\Delta C_{MCI^i} + \Delta C_{MPS^i} + \Delta C_{NP^i} \tag{5}$$

where $C^T{}_{total^i}$ and $C^t{}_{total^i}$ are the crop production in county *i* during the period *T* and *t*, respectively. $\Delta C_{yield^i}$ represents the contribution of yield to crop production changes in county *i*.

The degree to which each effect contributes to the change in the crop production of China's agriculture sector was estimated by the following equations. In this study, *T* = 2010 and *t* = 2000.

$$\Delta C_{X^i} \approx \sum_i L\left(C^T{}_i, C^t{}_i\right).ln\frac{X_i{}^T}{X_i{}^t} \tag{6}$$

where $L(a,b) = (a-b)/(ln\ a - ln\ b)$ is the logarithmic mean, $\Delta C_{X^i}$ denotes the crop production contribution of each factor (*X* : *yield, MCI, MPS*, and *NP*) in county *i*. The specific expression for each decomposition factor, given in LMDI form, is shown in Supplementary Data Table S1.

An overview of the entire methodological approach can be found in Figure 3.

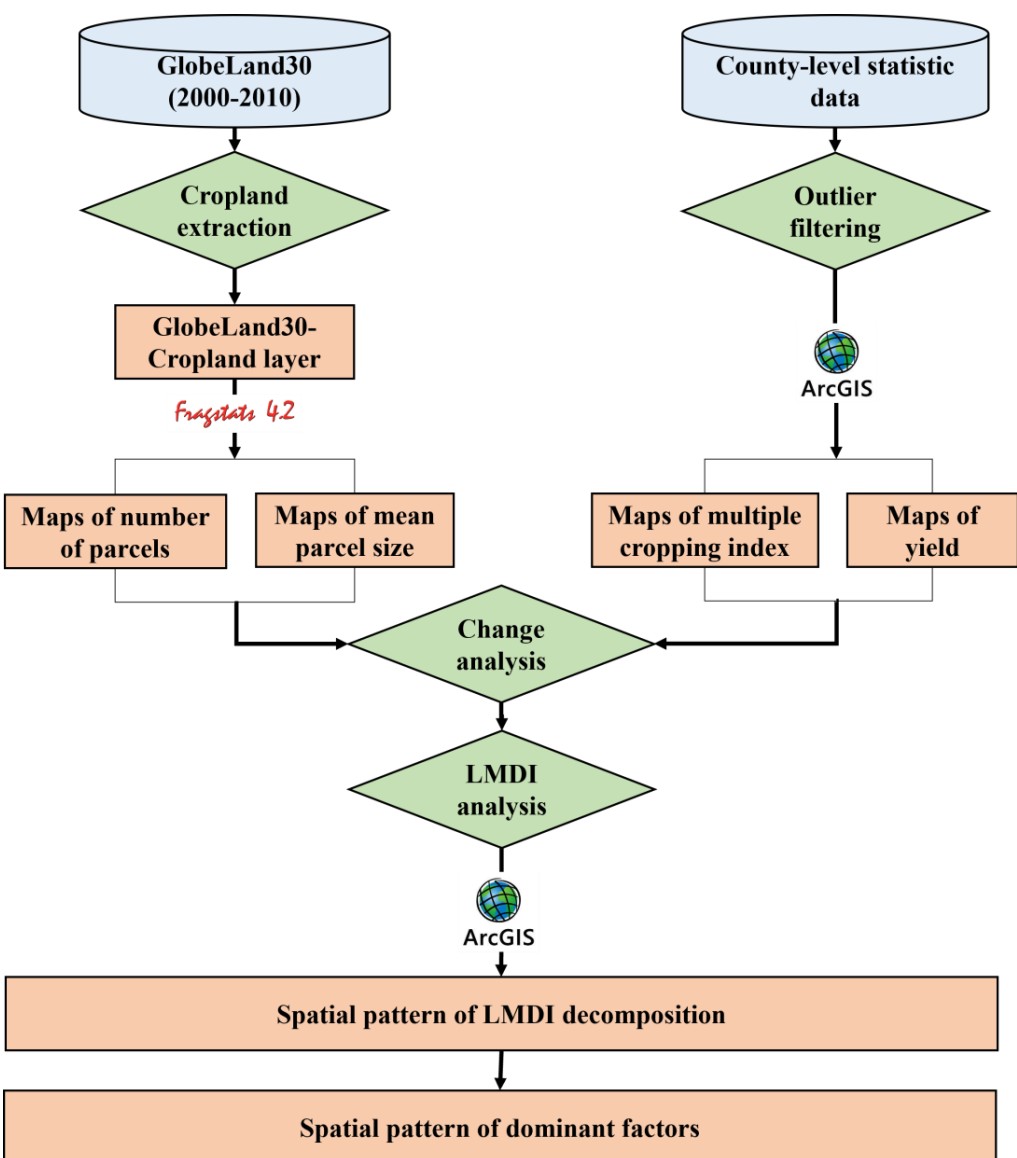

**Figure 3.** Flowchart of the methodological approach. Our study approach comprises the following steps: (1) Extracting cropland layer from GlobeLand30 and calculating landscape metrics using FRAGSTATS 4.2; (2) Calculating crop yield and multiple cropping index at the county scale based on statistical data; (3) Quantifying the contributions of all four indicators to crop production change, including food, feed, and fiber crops (cylinders = datasets, diamonds = data-processing and method, rectangular = results and maps).

## 3. Results

### 3.1. Spatial Patterns of Yield, MCI, MPS, and NP

The results show that yield and *MCI* are characterized by an increasing trend in most counties (Figure 4). Increases in yield were most pronounced in the Yangtze River Basin, the North China Plain and the northeast of China, which are the major crop production regions and are dominated by crops, such as rice, corn, and soybean. Together these regions account for 69% of national food production (National Bureau of Statistics of China, NBSC). These regions were also characterized by an increase in *MCI*. However, yield and *MCI* decreased across the southeast of China (Figure 4). This was especially the case along the southeast coast (i.e., Guangdong province, Fujian province, and Zhejiang province). When looking to the county scale, this study indicates that 62% of the counties (1495 out of 2396 counties) were characterized by an increasing trend in yield, whereas 31% had a decreasing

trend in yield. For *MCI*, 54% and 36% of counties were characterized by an increasing and decreasing trend, respectively.

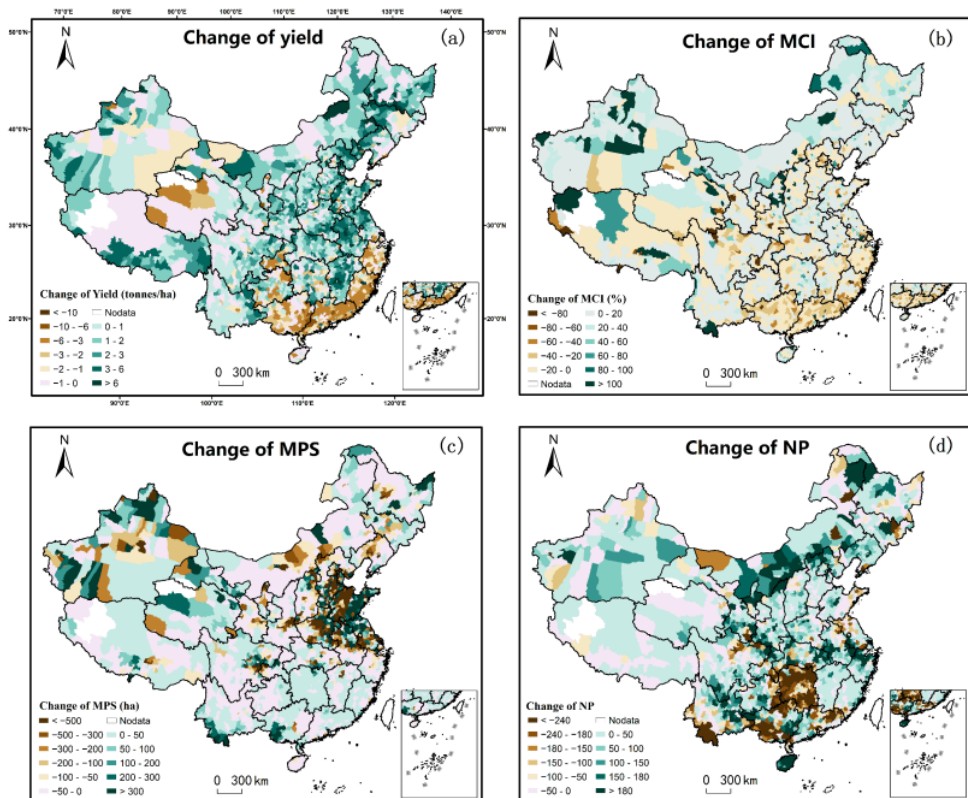

**Figure 4.** Characterization of change in four factors ((**a**): yield, (**b**): multiple cropping index (*MCI*), (**c**): mean parcel size (*MPS*), (**d**): number of parcels (NP)) at the county scale across China between 2000 and 2010.

The maps indicated an increasing trend in NP across most (61%) of the counties; for instance, there was a clear increasing trend in NP in the northern part of Inner Mongolia. Moreover, Jiangxi province and Zhejiang province were also characterized by an increasing NP. However, NP decreased in some counties in the central and southern part of China (e.g., in Hunan province and Guangxi province), indicating that agricultural parcels either became more fragmented or that cropland farming was disorganized. In contrast to NP, *MPS* was characterized by a general negative trend. More specifically, the results show a clear downward trend in *MPS* across the eastern part of China, which may be related to the conversion of croplands into other land cover types or a higher degree of landscape fragmentation.

### 3.2. Spatial Pattern of Crop Production Changes and National Decomposition

Since the 2000s, when China began to reinforce rural economic reform, crop production has been characterized by remarkable growth. This is reflected in the general increasing trend in crop production across the nation, as shown in Figure 5. Over the period 2000–2010, China's total crop production has increased by 30%. As our results indicate, more than 60% of the counties experienced an increase in crop production, which were mainly located in the northeast of China as well as across the North China Plain, also called "the breadbasket of China". A total of 70% of the area of these regions is characterized by flat topography (e.g., extensive plateaus or valleys), and therefore these are the predominate crop production areas making a significant contribution to national overall crop production growth. However, 29% of the counties across China were characterized by a loss in crop production,

which were mainly located in southeast coastal regions, scattered across the Guangdong, Fujian, Zhejiang, and Guangxi provinces.

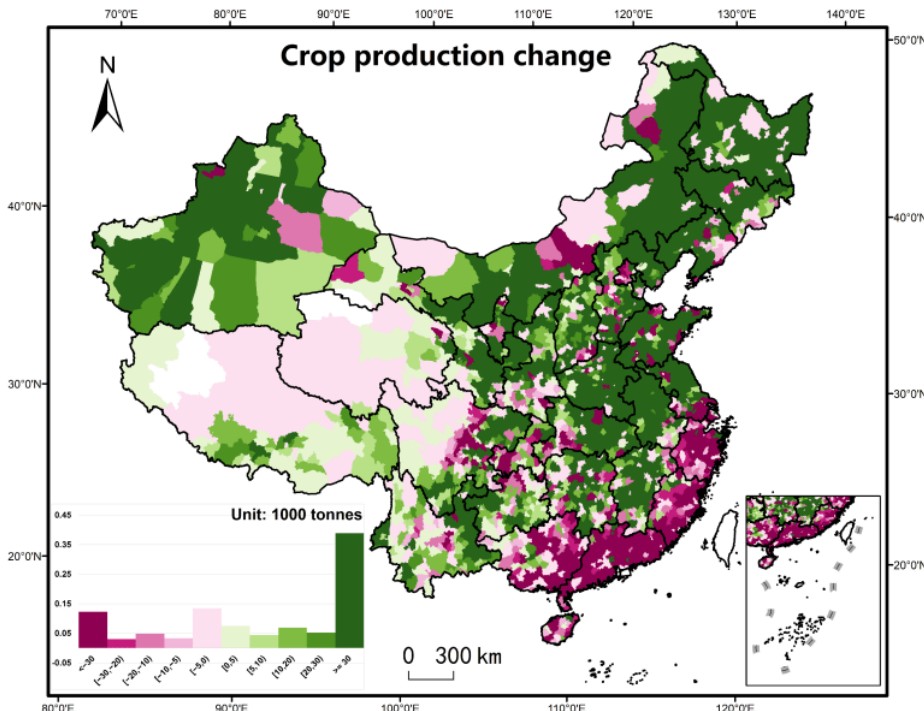

**Figure 5.** Map of crop production change at the county scale across China. The bar chart shows the proportion of the counties characterized by a given change in crop production corresponding to the intervals displayed in the legend of the map.

In the present study, the factor decomposition breaks the nationwide increase in crop production down into four contributing factors (i.e., yield, *MCI*, NP, and NPS). Figure 6 shows that the overall crop production increase in China has mainly been driven by three factors, i.e., *MCI*, yield, and NP, of which yield had the most important influence. More specifically, yield had a relative contribution of 77% to the total crop production change, whereas *MCI* and NP contributed only +27% and −21%, respectively. In contrast, *MPS* had a somewhat negative impact on crop production change, accounting for −25%.

### 3.3. County-Scale Decomposition

Figure 7 shows the county-scale decomposition map per factor (i.e., *yield, MCI, MPS*, and *NP* presented in the subpanels a-d, respectively) allowing us to obtain deeper insights into the spatial pattern of the various drivers changing China's crop production. This result shows that the increase in yield had a positive effect on crop production within 62% of the counties (1486 out of 2396 counties, Figure 7a). Among them, 43% of the counties (641 of these 1486 counties) have a contribution of more than 100,000 tons per county and 26% of more than 200,000 tons. These counties are mainly located across the major crop production regions (e.g., the North China Plain and the northeast of China). On the other hand, in 31% of counties (739 out of 2396 counties) yield had a negative influence on crop production; these are mainly located along the southeast coast of China and somewhat scattered in the west of China. However, among these 739 counties, less than 15% experienced a negative contribution of more than 100,000 tons, and most counties are restricted to a negative contribution of 50,000 tons.

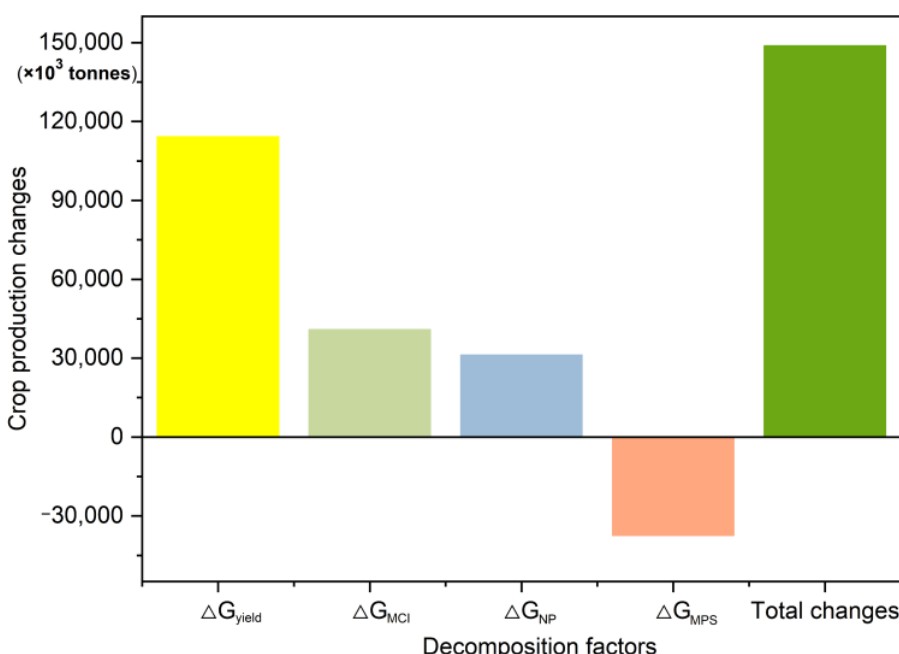

**Figure 6.** Factor decomposition of the total change in crop production across China between 2000 and 2010, considering the contributions of yield ($\Delta G_{yield}$), multiple cropping index (*MCI*, $\Delta G_{MCI}$), number of parcels (NP, $\Delta G_{NP}$), and mean parcel size (*MPS*, $\Delta G_{MPS}$).

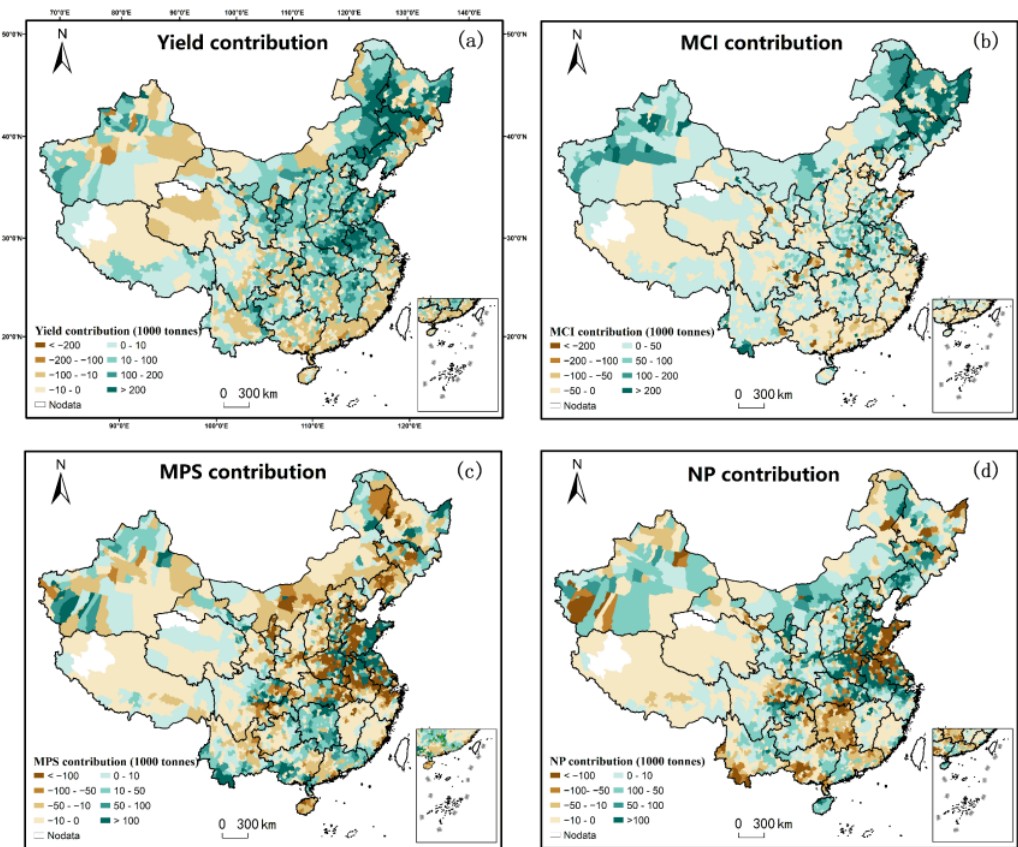

**Figure 7.** Decomposition of changes in China's crop production at the county scale with the individual contributions of the factors being given in the subpanels, (**a**): Yield, (**b**): Multiple Cropping Index (*MCI*), (**c**): Mean Parcel Size (*MPS*), (**d**): Number of Parcels (NP).

The results indicate that *MCI* decomposition has a positive effect on crop production within 54% of the counties (Figure 7b). Among them, 67% of these counties have a contribution of less than 50,000 tons. The analysis shows that these counties were concentrated in the provinces of Yunnan, Guizhou, and Hunan. However, in 38% of the counties, *MCI* had a negative influence on crop production; these are mainly located in the southeast of China and the North China Plain. The majority of these counties characterized by a negative contribution of *MCI* to crop production are limited to 50,000 tons.

Furthermore, *MPS* had a positive effect on crop production in only 37% of the counties (Figure 7c). Counties characterized by a remarkably positive contribution of *MPS* to crop production are located in important cropping areas across China, including the Chengdu Plain, Hunan province, and Shandong province. However, crop production was negatively impacted by *MPS* in 53% of the counties, which posed a threat to local food security. Nearly a fifth of the counties are characterized by a reduction of more than 100,000 tons of crop production due to the negative impact of *MPS*; these are mainly located in the provinces of Henan, Shandong, and Anhui.

Crop production was positively influenced by NP in 52% of the counties (Figure 7d), which were mainly located in the North China Plain, including Hebei province, Shanxi province, the Loess Plateau, and the southwest of China. This effect is limited, with 70% of counties having a contribution of less than 50,000 tons. However, some counties in Hunan province, the south of Yunnan, and the north of Jiangsu experienced a rather strong negative contribution of NP to crop production. In these counties, the overall negative contribution of NP to crop production is as great as $6311 \times 10^4$ tons, which should be considered as a substantial offsetting of the total NP-induced crop production increase (i.e., $3116 \times 10^4$ tons), as shown in Figure 6.

*3.4. Identification of Dominant Factors*

Comparing the decomposition analyses of the different factors could facilitate our understanding of the mechanism behind the observed changes in food provision at the national scale and provide us with useful information for policymaking. The map in Figure 8 presents the dominant factor influencing the crop production changes at the county level. This map also indicates whether there is a net increase (+) or decrease (−) in crop production. From this analysis we can see that in a large number of the counties where crop production increased, yield was the most dominant factor (i.e., 54% of the counties with increasing crop production), which were predominantly distributed across the northeast of China, in the provinces of Hebei, Shanxi, and Jiangxi. *MCI* was the second most dominant factor in counties characterized by increasing crop production. This accounted for 24% of these counties, mainly distributed across the northeast of China, such as Heilongjiang province. The proportion of counties characterized by an increase in cropland production in which *MPS* and NP were the dominant factors was relatively small and scattered across the nation, i.e., 13% and 14%, respectively,

Conversely, when considering the counties characterized by a decrease in crop production, approximately 33% of them experienced a dominant *MCI*-based impact. This is especially the case for the counties located along the southeast coast of China in which a remarkable decline in *MCI* caused a decrease in crop production. In 28% of the counties characterized by a decrease in crop production, this was the result of lower yields, which were spatially scattered. However, when comparing regions with either a yield-based negative contribution or yield-based positive contribution, there is still a clear overall net positive effect of yield on overall crop production. The counties with *MPS* and NP as the dominant factors leading a decline in crop production are scattered across China, accounting for 24% and 15% of the counties characterized by decreasing crop production, respectively.

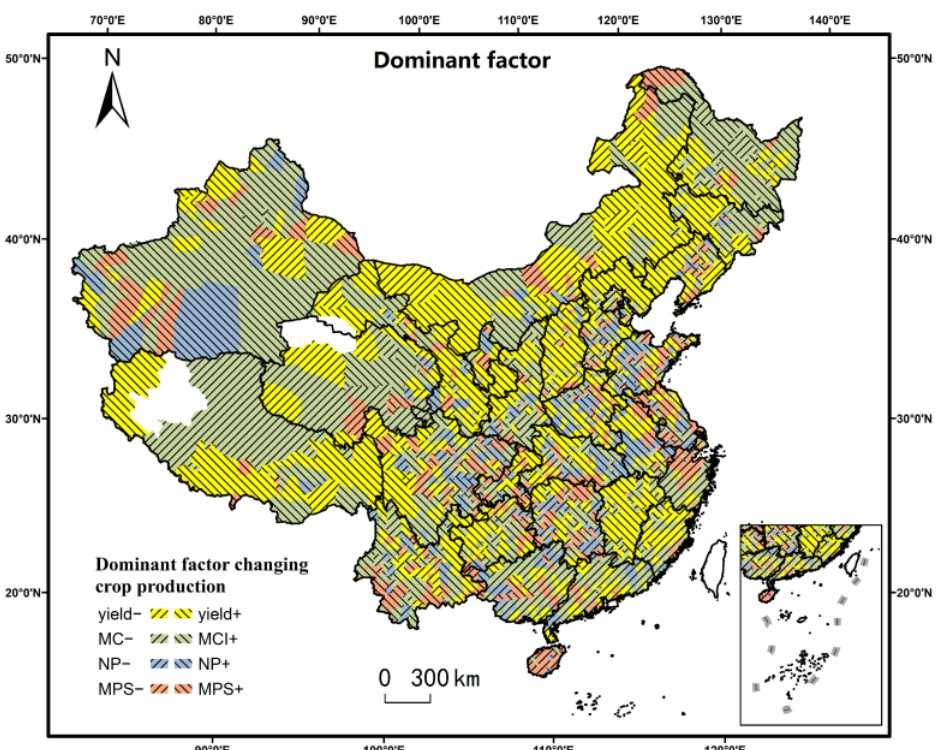

**Figure 8.** Spatial pattern of the dominant factors influencing crop production in China. Colors represent the dominant factor, whereas hatching indicates the overall crop production trend (i.e., "+" is net increase and "−" is net decrease in crop production).

## 4. Discussion

### 4.1. Crop Production Drivers

To our knowledge, this is the first attempt to conduct a detailed factor decomposition analysis of crop production in China at the county scale across the entire nation. In comparison with other studies investigating crop production trends, we found some similarities in terms of overall results but this study is characterized by some important methodological advances. For example, Li et al. (2016) decomposed grain production into yield, cropland area, and crop-mix at the (sub)national scale, whereas we were able to perform this research at the county scale [17]. They found that China's grain output increase between 1998 and 2013 was mainly driven by yield (+72%), which is similar to our results (+77%). However, their analysis did not consider the effect of changes in *MCI* on grain production increase, which was shown to be a key factor in our study. Other studies also underline the wide-ranging impact of *MCI* on agricultural production [40,41]. Zhou et al. (2015) found that yield (+101%) and *MCI* (+81%) had a positive effect on grain production between 1992 and 2012 [16]. However, this study is more limited than ours as it only analyses national scale changes due to the use of a coarse statistical dataset. In contrast, our novel approach decomposes the major factors influencing crop production change and enables the detection of more detailed spatial patterns at the county scale, which could be useful to support the development of regional crop policies.

To date, no studies have considered NP and *MPS* as contributors to crop production change. These factors seem to be key indicators for the degree of large-scale farming [42] and, hence, may determine land use patterns at the national/subnational scale. Consequently, we included these factors in our study. Our results suggest that area decomposition can provide a better understanding of how large-scale farming may affect crop production, therefore we believe that these factors should be taken into consideration when providing policymakers with science-based advice. In essence, considering the four factors used in this study (i.e., yield, *MCI*, NP, *MPS*)—combinations of which comprehensively reflect

the farming system—allows us to obtain substantially better insights into the mechanisms driving the observed crop production changes across China.

To our knowledge, few papers have rigorously quantified the effect of large-scale farming operations on crop production [14,43]. In essence, these studies found that land consolidation or land transfer due to large-scale farming increased the agricultural output (e.g., income). Although this can be seen as a significant conceptual and methodological advancement, our approach has some additional novelty because we applied the LMDI method instead of the typically used regression modeling approach.

### 4.2. Regional Diversity in Intensive Farming and Large-Scale Farming

Our research showed that both yield and *MCI* increased at the national scale, which indicated significant agricultural development across China. However, the high variability in these factors highlights considerable regional differences. This result has been confirmed in previous studies based on remote sensing and statical data [44,45], explaining that the high variability could well be due to climate change. More precisely, climate change has altered the cropping pattern across China [46,47], with some land becoming degraded and other land that was previously unsuitable for farming being cultivated. For example, rice cropping has moved northwards [48]. Furthermore, agricultural policies have helped farmers by encouraging farming incentives and revenues, which have contributed to the process of agricultural intensification [49]. However, the migration of young people from rural to urban areas may have resulted in a decline in yield and *MCI* [50]. This phenomenon is most present in the southern part of China (e.g., Chongqing municipality, Guizhou province, and Guangxi province). In the southeastern coastal areas (e.g., Guangdong province, Fujian province, and Zhejiang province) rapid urbanization has led to the conversion of vast cropland areas into artificial surfaces, which is probably the most important driver for yield and *MCI* decreases in these regions [51].

Additionally, we noticed an overall increase in *NP* and decrease in *MPS*, indicating that China is experiencing cropland fragmentation. This result was verified by previous studies [52,53], but the regional NP and *MPS* trends may diverge from those observed at the national scale. Furthermore, our findings indicate that the most important factor causing a decline in *MPS* across some counties in the North China Plain may be urbanization. This suggests that the current urbanization process may be sacrificing cropland and further fragmenting the larger cropland patches into smaller pieces [54]. Furthermore, policy support and agricultural innovation have affected the degree of large-scale farming, causing an increase in *MPS*. In the North China Plain, for example, *MPS* in some counties may have increased due to the land transfer policy, mechanization, and improved irrigation techniques. However, it should also be highlighted that some counties in the western and southern parts of China have been characterized by a general trend of agricultural land concentration due to the implementation of ecological restoration programs, e.g., the "Grain for Green" program, which have converted marginal cropland on sloping land into natural vegetation, decreasing the number of cropland patches [55].

### 4.3. Impact of Agricultural and Land Management Practices

When considering the overall increase in crop production, the shift towards crops characterized by higher-yield production seems to be an important driver [17,56]. Previous studies have indicated that yield was the largest driving factor for rising agricultural production across China between 1961 and 2009 [57]. The literature reveals that planting variety and planting structure benefit yield. For example, planting a mixture of rice varieties was effective in boosting yields [58,59]. Consequently, this has been widely promoted in China and increased the yield on average by 0.7 tons/ha, resulting in an income increase of 259 million US$. The amount of rice and corn planted has significantly increased as a result of the minimum purchase price for crops, the cost of temporary storage, and the high demand for livestock products [60,61]. A previous study found that the minimum purchase price is a key factor in the northeast of China, Jiangxi province, Anhui province,

and Jiangsu province, which may extend the positive contribution of yields in these regions, and therefore, outweigh other factors [62].

However, yield is not the only factor increasing crop production. Our study also found that an increase in *MCI* had a positive effect on crop production, but its impact was considerably smaller [57,63]. Specifically, our findings demonstrate that *MCI* changes have a direct impact on food production through planting management and land restructuring. For example, the increase in farmers' incomes actively encourages the scientific management of cropland and enhances intensive farming and crop production [64]. As the profitability of staple crops declined from 2004 onwards (NSBC, 2005), crops with higher production and higher *MCI* (e.g., vegetables) replaced rice crops [65,66], which resulted in a change of cropping structure [67]. However, this is not the case in all regions. Our analysis using county-scale decomposition shows that different degrees of farming intensification have different effects on rising crop production during the first 10 years of the 21st century. Some counties located in the North China Plain had a positive impact of crop production, which was due to the use of agricultural inputs (e.g., fertilizers, pesticides, etc.) as well as improved irrigation methods resulting in an increase of both yield and *MCI* [68,69]. The increase in the *MCI* in Henan province has contributed to the increase in crop production, which can be explained by the expansion of vegetable cultivation [70]. Despite the fact that counties located along the southeastern coastal area tend to have a more favorable climate for agriculture and stronger economic development, they are typically characterized by a negative contribution to overall crop production due to intensive urbanization.

Our results also showed that while the growth of NP represents an increase in crop production, the drop in the national *MPS* had a negative impact on the increase in crop production at a national scale. This is undeniable given that the rapid decrease in *MPS* and gain in NP are a direct result of increasing cropland fragmentation across China [52]. Although haphazard and unplanned cultivation partly increased the cropland NP (i.e., recultivation mainly occurred in mountain areas due to conversion of grassland and woodland to cropland) [71], it was unable to compensate for the loss in crop production due to the sharp decline in *MPS* as a result of the cropland area lost within the given period. When we look more closely at the NP and *MPS* impacts at the county scale, we discover important inter-regional differences. As observed, the shrinkage in cropland size had a harmful effect on food security in the North China Plain and in the northeast of China, where the terrain is flat and mechanized agriculture is widely established. Rapidly expanding urbanization in those areas resulted in a conversion of large-scale cropland to more scattered patches. There are more patches and obviously more plots for agriculture, but the mean patch area per cropland is declining. This means that more agricultural inputs (e.g., fertilizers and pesticides) will be needed to produce equal crop output [72,73]. It has been proven that more efficient planting management (e.g., after joining multiple small cropland patches into larger ones) results in a more rational allocation of agricultural resources [74]. This may result in higher overall crop production. For example, farmers can lower their costs, and hence maximize their agricultural output, by making more use of scientific insights into soil quality related parameters. Furthermore, Mekki, et al. [75] found that farmland fragmentation significantly affected the decision-making process with regard to crop allocation. This is particularly true for the southern part of China where staple crops (e.g., maize, rice) do not support the income of local farmers and as such have been replaced by crops with a higher economic return (i.e., cash crops) or in extreme cases croplands have been abandoned [76,77]. The shift to more fragmented croplands in staple crop-producing areas indicates that this factor could increasingly offset gains from improved yield and *MCI*.

### 4.4. Implications, Limitations, and Future Research Avenues

Important conclusions can be drawn from our study in order to support policymaking with the objective to meet future food demand by making optimal use of the existing cropland area within China. Our results show that while crop production continued to rise, the relative importance of the drivers behind the observed overall crop production

varied across the regions. In essence, yield and *MCI* were the predominant positive factors contributing to crop production increase during the first decade of the 21st century, whereas *MPS* had a negative impact. However, the importance of these main factors driving crop production trends may change considerably in the near future. This underlines the need for continuous research to identify the drivers of crop production changes.

A key question for enhancing future food security will be how to further improve crop production [78,79]. However, based on our findings we can make the following recommendations.

First, yield growth has slowed over the past 20 to 30 years, and even stagnated in some important food-producing regions, possibly because the current crop varieties are reaching their biophysical limits [80]. By accounting for *MCI* potential and land consolidation, counties that primarily relied on yield-driven production, such as Shanxi province and Jiangxi province, can boost crop production while preserving the current yield growth.

Second, priority should be given to improving *MCI* because the counties that relied on other factors to increase crop production are mostly located in the southwestern mountainous part of China, where there is still a lot of potential to further increase *MCI* [31,44,49]. Consequently, an interesting additional financial support could be the provision of credit and insurance products for key infrastructure, such as advanced irrigation techniques in order to allow a further increase of *MCI* [31,81,82].

Third, NP and *MPS* have an obvious effect on crop production in regions, such as the North China Plain and Xinjiang province, because of the large-scale farming in these areas. Hence, considering the fertile soils and flat topography in these regions, we suggest to continue increasing the size of the fields through land consolidation as well as providing financial support in order to set up service centers enabling land transfer and mechanization, which benefit green production [83,84] and therefore, "Ensuring Sustainable Consumption and Production (SDG12)". The "Requisition–Compensation Balance Project", implemented in 1998 [85,86], can also effectively increase land productivity as it ensures that the total land use balance remains stable, preventing the fragmentation of cropland and the abandonment of fertile land [87].

It is important to recognize that our approach has several limitations. However, by assessing these limitations we were able to identify some important avenues for future research:

First, the crop production data used in this study does not contain specific crop type information. However, changes in crop type composition can affect crop production in the sense that a shift towards higher-yielding crops would result in higher overall crop production without any crop type being characterized by a significant improvement in yield [1]. Hence, the lack of detailed data limits the ability to identify key crop types and assess their contribution to driving crop production in a spatially explicit manner. For example, *MCI* is given per county as an averaged value for all crop types due to the unavailability of crop type specific data. An interesting future study could be to apply a GIS analysis (e.g., adopting a Spatial Allocation Model [88]) using specific crop type data in order to separately obtain the spatial distribution of each crop type. This may facilitate our understanding regarding the impact of proportional changes in crop types on crop production dynamics across different spatio-temporal scales.

Second, this study only covered two points in time, 2000 and 2010. This could be an issue as crop production tends to be more stable within a relatively short period of time, and hence changing phases are easily missed when analyzing at decadal time steps. So, this kind of research may require a longer time period to better identify the long-term trends. However, the recent remarkable increase in crop production has also experienced some fluctuations, and in-depth analyses using temporally more detailed datasets are required in order to obtain a better understanding of inter-annual fluctuations and the reasons behind them.

Third, statistical data can be unreliable. For example, the information regarding the cropland surface is particularly uncertain. Statisticians have overestimated yield before 2006 in order to compensate for the underreported land area [89,90]. On the other hand, after 2006, farmers and local governments have often overestimated their land area in

order to receive more subsidies [91], and that way, have had to reduce the reported yield in order to meet the total production. Consequently, as the reported yield values were lower than the true yield values, our results may underestimate the contribution of yield and overestimate the contribution of *MCI*. To overcome this issue, the use of remote sensing derived proxies for yield, such as NDVI, could be beneficial [92,93]. The contributions from NP and *MPS* are not affected by the statistical data, since they were retrieved from GlobeLand30 satellite data.

Altogether, the present factor decomposition of crop production facilitates the identification of future constraints and opportunities for the Chinese agricultural sector. Until now, science and technology have had a particular focus on increasing food through intensifying agriculture practices, resulting in yield and *MCI* increases, in order to avoid an expansion of cropland area. However, this study highlights that besides intensifying agriculture practices, large-scale farming also has the potential to boost crop production in many regions across China. In a nutshell, a better understanding of the drivers of change in crop production can suggest strategies for policymakers to address the SDGs focused on enhancing future food security.

## 5. Conclusions

This study shows that yield, *MCI*, NP, and *MPS* are key factors affecting crop production changes across China between 2000 and 2010. However, the results highlight that the spatial distributions of the contribution of these factors were remarkably different. By applying a LMDI approach, we found that yield had the most important positive effect on crop production increase with a relative contribution of +77%. In addition, *MCI* (+27%) and NP (+21%) had a positive effect on crop production changes, whereas *MPS* had a negative impact. The counties characterized by a positive effect of yield accounted for 62% of the total number of counties, which were mainly situated in the north of Hebei and Shanxi provinces. The counties where *MCI* had a positive effect on crop production increase accounted for 54% of the total number of counties, which were mainly located in Heilongjiang province. *MPS* and NP had a positive impact on crop production increase in 52% and 37% of the total number of counties, respectively, which were mainly located in the North China Plain. Despite the overall increasing trend in crop production, some counties were characterized by a decrease and therefore require special attention. When looking into the areas characterized by a decrease in crop production, a decline in *MCI* and yield had a primary negative effect on crop production, accounting for 33% and 28% of the counties with decreasing crop production, respectively. Furthermore, *MPS* and NP were the main factors associated with a decrease in crop production in counties scattered across China, representing 24% and 15% of the counties with decreasing crop production, respectively. The negative contributions to cropland production increase were mostly due to a decline in large scale farming. Besides stabilizing yield and maintaining the total NP, land consolidation and financial assistance for land transfer and advanced infrastructure are critical strategies that could help to counter the fragmentation of croplands and *MCI*, and therefore make an important contribution to the sustainable development of agriculture in China.

**Supplementary Materials:** The following supporting information can be downloaded at: https://www.mdpi.com/article/10.3390/rs14246399/s1, Figure S1. Cropland area in GlobeLand30 and statistical data correlations for 2000 and 2010. (a) the correlations in 2000. (b) the correlations in 2010. Each point presents the data of a county, in total 2396 counties were computed. Linear regression was used to measure the $R^2$ with the red fitting line. Table S1. Decomposition factors for a given county, mean parcel size, and number of parcels are calculated based on FRAGSTATS 4.2 software. $yield_i^{2010}$ represents yield in 2010 in county $i$.



**Author Contributions:** Conceptualization, Y.L.; Methodology, Y.L.; Software, Y.L.; Writing—original draft, Y.L.; Writing—review & editing, J.W. and J.M.; Visualization, Y.L. and J.M; Supervision, W.W., J.W., G.C. and J.M.; Project administration, W.W. and J.M. All authors have read and agreed to the published version of the manuscript.

**Funding:** This work is financed by the National Natural Science Foundation of China (41871356).

**Data Availability Statement:** Not applicable.

**Acknowledgments:** We would like to acknowledge the anonymous reviewers and editors whose thoughtful comments helped us to improve this manuscript.

**Conflicts of Interest:** The authors declare no conflict of interest.

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
