# Peer review of "An In-Depth Assessment of the Drivers Changing China’s Crop Production Using an LMDI Decomposition Approach"

_remotesensing, doi:10.3390/rs14246399_

Round 1

Reviewer 1 Report (Previous Reviewer 1)

The mapping of the South China Sea is still non-standard. The nine-dash line has been adjusted to ten-dash line.

Line 228, The fomat of the variables in the formulas must be same as in the following introduction text. For example i would be italic. 

Author Response

The mapping of the South China Sea is still non-standard. The nine-dash line has been adjusted to ten-dash line.

Response 1: Thanks for your suggestions. We have redesigned all figures and added “The ten-dashed Line”.

Line 228, The format of the variables in the formulas must be same as in the following introduction text. For example i would be italic.

Response 2: Thanks for your suggestions. We revised the variables in the formulas and corresponding text in order to make sure that they are the same.

Reviewer 2 Report (Previous Reviewer 5)

General comments:

The manuscript, entitled "An in-depth assessment of the drivers changing China’s crop production using an LMDI decomposition approach” focuses on the assessment of the drivers changing China’s crop production using an LMDI decomposition approach. There are some issues with this manuscript, mainly related to the readability and composition of the manuscript.

Please review the quality of your English throughout the manuscript.

Specific comments:

Point 1: No need of writing in the abstract section…Although our results are consistent with findings from previous studies…this kind should be written in the discussion section

Point 2: Figure 6; the bar graph is not attractive to the readers, so I suggest that modify the bar graph.

Author Response

No need of writing in the abstract section…Although our results are consistent with findings from previous studies…this kind should be written in the discussion section.

Response 1: Thanks for your suggestion. We decided to delete the sentence. However, we have put a similar sentence in the “discussion” section (Please see line 374-377).

Figure 6; the bar graph is not attractive to the readers, so I suggest that modify the bar graph

Response 2: Thanks for your suggestion. We made the bar graph more attractive by using colors. Please note that the color of each bar are consistent with the decomposition result shown in Fig8.

This manuscript is a resubmission of an earlier submission. The following is a list of the peer review reports and author responses from that submission.

Round 1

Reviewer 1 Report

Line 185, The South China Sea is incomplete in Figure 2.

Line 99, The objectives of this research are: (1)How do some factors? (2), (3)

Line 105, (i) changes into (1).

Line 110, The figures are unclear. The authors just need to give the items' names.

Line 264, The grain production is mainly affected by the planting structure,  planting varieties, planting management level. The authors need make more discussion on these items.

How to validate the decomposition results in this study?

Reviewer 2 Report

The topic of this study is interesting. However, it does not bring enough novelty in terms of remote sensing to justify publication in REMOTE SENSING. The RS methods the authors use are not very innovative, and therefore I recommend the authors submit it to another journal with a closer orientation to the main topic of interest. Therefore, this paper would be more appropriate for a journal dealing with agriculture or agronomy. Before submitting this paper to another journal, I would suggest focusing on the main topic and making the methodology clearer. The writing of this manuscript needs to be improved. The authors should clearly emphasize the strengths of their study/theory/methods.

Reviewer 3 Report

Review of “An in-depth assessment of the drivers changing China’s grain production using a LMDI decomposition approach”

Journal: Remote Sensing

Manuscript Number: remotesensing-1922631

Author: Yuqiao Long, Wenbin Wu, Joost Wellens, Gilles Colinet and Jeroen Meersmans

The research explains a critical scientific question about the food production and critically analysed using the data and methods. It is interesting research and well-written.  The analysis done and the graphical presentation explains the idea well.

I would accept the publication and recommend to publish in the Remote Sensing journal.

Reviewer 4 Report

This manuscript analyzed grain production trend at a national county scale. Globeland30 and statistic data were involved and corresponding driving factors were taken into consideration, which is of a good idea to make a large-scale analysis. This makes the whole work to have scientific meanings as well as practical one. Nevertheless, I listed some recommendations and comments as follow for an improvement.

1. The research objectives are not well-established due to the relatively vague introduction. The authors stated that “large set of factors may have impacts on agricultural production change” (line 54-68), while when to solve this problem in this study, the authors tried to use some LU data-derived factors (i.e. MCI, NP, and MPS). However, it is really hard to solve the existed uncertainty of those factors caused by LU data accuracy, in other words, of globeland30 data has uncertainty itself, how to guarantee the analysis results?

2. I feel a little bit confused that how to make factor calculating in year, since as I understand, Globeland30 only have data in the year of 2000, 2005 and 2010 (actually even derived from different remote sensing data).

3. From the results it is very valuable to see the spatial distribution/pattern of different selected factors, while a deep analysis with more insights are relatively lacking.

6. There are also many formatting problems that I see the original manuscript did not use MDPI template so well.

Reviewer 5 Report

General comments:

The manuscript, entitled "An in-depth assessment of the drivers changing China’s grain production using LMDI decomposition approach” focuses on the assessment of the drivers changing China’s grain production using LMDI decomposition approach. There are some issues with this manuscript, mainly related to the readability and composition of the manuscript.

Please review the quality of your English throughout the manuscript.

I would like to see some recommendations of this study to policymakers and implementers at the end of the abstract section.

Specific comments:

Point 1: Line 41 to Line 43 China has…United Nations…correct the English and write source.

Point 2: Line 49…impacts (write source).

Point 3: Line 53…China (Write source).

Point 4: Line 59…production (write source).

Point 5: Line 75…production (write source).

Point 6: Line 83…scales (write source).

Point 7: Line 111. Figure 1; please make it visible to the readers. Do it again with the grid, legend, scale, and north arrow. For the figures also write the title of maps in the appropriate font size.

Point 8: Line 231. Figure 4; lacks visibility and do it again according to cartographic standards. Use appropriate font size for the legend, scale grid, etc.

Point 9: Discussion section (Line 264), this is an important part of the study and I suggest to the authors compare your results with other similar studies in order to emphasize the originality and novelty of the study.

Point 10: Line 404. I suggest that write recommendations at the end of the conclusion section or separately write recommendations to the policymakers.

Point 11: Line 405: References: The authors used only 31 citations (references) it is so small for the article please add more citations in the introduction and discussion section.